# Effects of Silicone Oil Viscosity and Carbonyl Iron Particle Weight Fraction and Size on Yield Stress for Magnetorheological Grease Based on a New Preparation Technique

**DOI:** 10.3390/ma12111778

**Published:** 2019-05-31

**Authors:** Kejie Wang, Xiaomin Dong, Junli Li, Kaiyuan Shi, Keju Li

**Affiliations:** State Key Laboratory of Mechanical Transmission, Chongqing University, Chongqing 400044, China; wangkejie@cqu.edu.cn (K.W.); junlili@cqu.edu.cn (J.L.); kaiyshi0@gmail.com (K.S.); likjuc@gmail.com (K.L.)

**Keywords:** magnetorheological grease, yield stress, silicone oil viscosity, carbonyl iron particle fraction, carbonyl iron particle size, constitutive model

## Abstract

This paper investigated the effects of silicone oil viscosity (SOV) and carbonyl iron particle (CIP) weight fraction and size on dynamic yield stress for magnetorheological (MR) grease. The MR grease samples were prepared using orthogonal array L_9_ on the basis of a new preparation technology. The shear rheological tests were undertaken using a rotational shear rheometer and yield stress was obtained based on the Bingham fluid model. It was found that CIP fractions ranging from 65 wt% to 75 wt% and SOV varying from 50 m^2^·s^−1^ to 1000 m^2^·s^−1^ significantly affect the magnetic field-dependent yield stress of MR grease, but the CIPs with sizes of 3.2–3.9 μm hardly had any influence based on the analysis of variance (ANOVA). In addition, the yield stress of MR grease mainly depended on the CIP fraction and SOV by comparing their percent contribution (PC). It was further confirmed that there were positive effects of CIP fraction and SOV on yield stress through response surface analysis (RSA). The results showed a high dynamic yield stress. It indicated that MR grease is an intelligent material candidate which can be applied to many different areas requiring high field-induced rheological capabilities without flow for suspension. Moreover, based upon the multivariate regression equation, a constitutive model was developed to express the function of the yield stress as the SOV and fraction of CIPs under the application of magnetic fields.

## 1. Introduction

Magnetorheological (MR) fluids are smart materials whose rheological properties can be changed significantly, rapidly, and adjustably under the application of magnetic fields [1]. MR fluid has attained widespread attention owing to its excellent magnetic field-induced rheological performances in many applications, such as clutches, brakes, dampers, and shock absorbers requiring active intelligent control for torque transmission or vibration [2,3]. Nevertheless, the phenomenon of settlement for MR fluid is generated in these devices due to the density mismatch between ferromagnetic particles and carrier fluids [4,5]. The sedimentation is typically difficult to redisperse and degrades the MR response, thus hindering the engineering applications of MR fluids [6]. In order to solve this settling in the MR fluid, various means have been developed; adding thxiotropic agents and surfactants to form a thixotropic network and enhance antisettling hydrodynamic effects [4,7,8,9]; coating the magnetic particles with polymers to decrease density [1]; and adding magnetic nanoparticles to produce steric repulsion between the micron-scale carbonyl iron particles (CIPs) [10]. Although many researchers have made great efforts in this area, the problem of settlement stability for MR fluids is not fully solved. Based on the method of changing the suspension medium, the grease, due to its viscoplastic characteristics, provides another possibility to conquer the problem and is a promising candidate as a suspension medium [5,11]. Therefore, in this work, grease is selected as an MR carrier medium to eliminate particle sedimentation.

Grease is a colloidal system consisting of a thickener of different materials in base oil [12]. The thickener has a three-dimensional fibrous structure to stably suspend magnetic particles in equilibrium against gravitational forces [13,14]. Grease has an apparent viscosity and thus cannot flow easily [12]. Thus, MR grease with good sedimentation stability and intrinsic viscosity is applicable for controllable damping systems, especially seismic dampers, which require leak avoidance [15]. The torque output of an MR fluid under the presence of magnetic fields is enhanced as the shear rate increases, thus the torque of an MR fluid strongly depends on the operational speed [16]. As the speed of an MR device increases, due to the speed-dependent torque transfer, the MR fluid might exceed the allowable torque limit. In contrast, MR grease can provide constant torque output to high-speed applications [16,17]. It is noteworthy that MR fluid is prone to leakage in MR devices [15,18]. Leakage may give rise to instability or failing of the equipment. To prevent MR fluid from leaking, the apparatus requires appropriate sealing, thus raising the manufacturing cost [18]. Unlike MR fluid, MR grease, due to its viscosity, has a significant sealing effect and has been applied to a shear-type dampers, thus avoiding the requirement of sealing elements [19,20]. Although grease is used as a suspension medium, leading to high off-state viscosity of the MR grease, the suspension medium does not significantly affect the MR response [5]. Furthermore, based on the statistical inference from the accelerated life test, the storage life span for MR grease under nominal temperature is forecasted to be 15.2 years [21]. This indicates that MR grease is advantageous for long-term storage stability, which exceeds the requirements of many industrial applications. It should be noted that the shear viscosity of MR grease typically decreases with increasing shear rate [11]. This behavior is known as shear thinning and is attributed to the inherent lubrication provided by grease between moving surfaces within high operation velocity ranges [15,19]. Furthermore, MR grease can decrease the loss rate of the lubricating grease and improve the lubrication to reduce friction [22].

As mentioned above, MR grease has many advantages, such as its antisedimentation ability in engineering applications. However, there are few studies investigating how the weight fractions of CIPs, sizes of CIPs, and silicone oil viscosity (SOV) jointly influence the field-dependent yield stress. The yield stress is important in the appropriate design of MR devices in terms of the torque output [8,16]. Ranking et al. [5] studied the use of grease as the carrier fluid for MR suspension and found that it is able to prevent sedimentation. To investigate the influence of grease on suspension, the authors observed the field-dependent rheology of MR greases with different concentrations of the grease medium. The results show that the viscoplastic medium with yield stress ranging from 0.9–37 Pa did not influence the steady shear MR response of the suspension. Furthermore, they claimed that the field-induced yield stress has a subquadratic relationship with the magnetic flux density. After that, Park et al. [23] explored the relationship between nanoadditives and the rheological response of MR grease. In [23], it was demonstrated that the dynamic yield stress of the suspension with the grease medium was not affected by nano-sized ferromagnetic CrO_2_ particles. Simultaneously, the synergistic effect of nanoadditives in the suspension did not reduce the yield stress at low magnetic field strength. To confirm whether different magnetic particle shapes affect the mechanical properties of the MR grease, Wei et al. [24] calculated the shear yield stress based on computer simulation. It was found that MR grease with nonspherical (hexagonal) magnetic particles can produce larger shear yield stress compared to spherical particles. Recently, Mohamad et al. [18] researched the weight percentage of CIPs in MR grease as a function of field-induced rheological properties at different magnetic flux densities. The results showed that the particle fraction and the on-state magnetic field could affect the dynamic yield stress of the MR grease. In [18], the authors achieved the highest yield stress (52.7 kPa) of the MR grease with a weight fraction of 70 wt% under a magnetic field of 0.851 T. In contrast with the study of above-mentioned Mohamad and co-workers, Kavlicoglu et al. [16] discussed the effect of CIP size (5–45 μm) on the torque performance of high-density MR grease with large particle fractions (90–99 wt%). Researchers tested the torque output of MR grease in an MR clutch. From this study, they showed that CIP size does not influence torque capability [16]. Meanwhile, it was discovered that a CIPs fraction of 90 wt% generated a shear yield stress value of 36.1 kPa for an input current of 2 A [16]. In addition, Sukhwani et al. [25] investigated the comparative braking property of MR grease and MR fluids at the high particle weight percentage of 90 wt%. The results indicated that MR grease has lower on-state brake torque output than MR fluids, which has hindered the industrial application of MR grease. Generally speaking, this literature review indicates that MR grease is a promising MR material that is widely used in many applications, such as seismic steering systems, damping systems, timing triggers, and so on. Previous research has mainly focused on how the viscoplastic medium, nanoadditives, shape of magnetic particles, and CIP size and fraction affect the field-induced rheological behavior of MR grease based on conventional preparation techniques. Although many investigations regarding MR grease have been carried out, how larger-span SOV, moderate weight fractions of CIPs and smaller CIP sizes) influence the field-dependent dynamic yield stress has not been clearly studied.

Therefore, the main technical contribution of the present work was to experimentally study the effects of different SOV, moderate fractions of CIPs, and different CIP sizes on the rheological performance of MR grease based on a new preparation technique. The function of shear rate and shear stress for MR grease was characterized under the action of magnetic fields varying from 0 to 0.7 T at 25 °C using a commercial rheometer. The yield stress values were determined based on the Bingham fluid model by calculating shear stress–shear rate curves to zero shear rate and searching for the point of intersection with the vertical axis. In order to clearly investigate effects of SOV, CIPs fraction, and CIP size on the yield stress for the MR grease, four analysis methods (analysis of variance, comparative analysis of contributions, response surface analysis, and rheological constitutive relation characterization) were applied in this study. Three input parameters (i.e., SOV, CIPs fraction, and CIP size) were qualitatively evaluated to determine whether they significantly affect the field-induced yield stress (response parameters) through analysis of variance (ANOVA) based on the Taguchi L_9_(3^3^) test program. The strength effects of the three input parameters on the yield stress were quantitatively evaluated by using the comparison of percent contribution (PC). The direction of discrepancy between input parameters (i.e., SOV and CIPs fraction) to yield stress were discussed via response surface analysis (RSA) on the basis of the multivariate regression equation. In addition, based on the multivariate regression equation, a field-dependent rheological constitutive model was proposed to characterize the yield stress as a function of the CIPs fraction, SOV, and magnetic field, and its prediction accuracy was discussed accordingly.

## 2. Materials and Methods

### 2.1. Materials

The spherical CIPs (MPS-MRF series) were purchased from Jiangsu Tianyi Ultra-fine metal powder Co. Ltd., Huaian, China. Silicone oils (PMX-200 series) were purchased from Dow Chemical Company, America. The antioxidant, T-203 was purchased from Jinzhou Snda Chemical Co. Ltd., Jinzhou, China. Lithium stearate was supplied from ASK Lubricants Co, Shenzhen, China. The lithium hydroxide was obtained from Fuchen Tianjin Chemistry preparation Co. Ltd., Tianjin, China, whereas the boric acid was provided by Shanghai Aibi Chemistry Preparation Co. Ltd., Shanghai, China.

### 2.2. Input Parameters and Response Parameters

The dynamic yield stress is the force required for the suspension flow to rupture the fibrous columns of magnetic particles aligned head-to-tail along the magnetic flux lines under the action of a magnetic field [26]. The yield stress is the maximum obtainable alterable damping for an MR damper and is critical in the design of MR devices [8,16,27]. Consequently, the yield stress under application of different magnetic field strengths (varying from 0 to 0.7 T) was used as the response parameter.

Park et al. [11] found that MR grease exhibits Bingham rheological behavior of yield stress in the absence of a magnetic field, due to the inherent ability of grease to forma magnetic particle chains. Mohamad et al. [18] also reported that chain deformation of magnetic particles might be affected by grease. Grease was thickened with silicone oils (base oils) and lithium soap. The silicone oil viscosity (SOV) has an important effect on the rheological properties of the grease medium [28]. Although research on the effect of different fractions of greases on the field-dependent yield stress has been conducted, there are few studies on how the SOV affects yield stress. Thus, the SOV was varied from 50 to 1000 m^2^·s^−1^ as one of the input parameters (test factors) in this study.

Based on the literature review, the results showed that the field-induced properties of MR grease were nearly independent of CIP size [16,27]. However, most studies typically use large 4–53 μm CIPs. It is still unclear whether CIP sizes below 4 μm affect the field-dependent yield stress of MR grease. Thus, CIP size, ranging from 3.2 to 3.9 μm, was used as another input parameter.

Most previous research has focused on the effect of particle fractions lower than 65 wt% and higher than 75 wt% on the magnetic field-dependent performance of MR grease. Studies on the contribution of moderate CIPs fractions compared to CIPs size and SOV towards the field-induced yield stress are much rarer. Consequently, moderate CIP fractions in the range 65–75 wt% was selected as one of the input parameters.

### 2.3. Experiment Design

Taguchi experiment design method [29,30] was applied in this work. This method adopts a particular series of arrays known as orthogonal arrays. These normal arrays allow conducting a minimal number of tests that supply complete information of all input parameters influencing the response parameter [29]. In order to understand the effects of the three input parameters (i.e., SOV, CIPs fraction, and CIP size), an L_9_(3^3^) orthogonal array is appropriate. Table 1 shows the three input parameters with each parameter having three level values. These input parameters were allocated under different columns as show in Table 2 and Table 3. In Table 1, the codes −1, 0, and 1 denote the low, central, and high levels, respectively.

### 2.4. Material Preparation

Typically, MR grease was prepared by directly mixing commercial greases with CIPs. Although these MR greases have good antisettling ability, their three-dimensional network structures may not be controlled. In order to manipulate the interior crosslinking structure between the grease medium and the CIPs, the CIPs were added during soap fiber growth. A new MR grease preparation method was used in this work, as shown in Figure 1. Using the L_9_ orthogonal array program, as shown in Table 2 and Table 3, a series of MR grease samples was synthesized with silicone oils, CIPs, lithium stearate, boric acid, lithium hydroxide, and an antioxidant. Initially, half of the silicone oils was mixed with 6.4 wt% lithium stearate in a beaker and the mixture was stirred for 20 min at 25 °C. Then, the boric acid solution and lithium hydroxide solution were slowly added to the above mixture in turn at 80 °C. After stirring the mixture for 10 min, saponification reaction was conducted for 30 min at 120 °C. When the reaction finished, the eutectic product of lithium stearate and lithium borate was initially synthesized. After that, the heating temperature was raised to 140 °C and maintained for about 150 min until the water was entirely evaporated. Meanwhile, the lithium soap (the above-mentioned eutectic product) was formed and the weight ratio of lithium borate to lithium stearate was 2:8. After evaporation, the other half of the silicone oils and the antioxidant at a weight fraction of 0.7 wt% were mixed into the lithium soap and the mixture was held at 140 °C. Ten minutes later, the grease was preliminarily formed. Then, the CIPs were mixed with the grease and the mixture was stirred for 10 min. The mixture was kept at 200 °C for 7 min to induce crosslinking of CIPs and the grease medium. Afterwards, the mixture was self-cooled for 10 min to allow formation of a tighter fibrous network of CIPs and grease. After uniformly grinding the mixture for 30 min, the desired MR grease was finally prepared. The appearance of MR grease samples after resting for 30 days is shown in Figure 2. There was no obvious sedimentation in these MR grease samples, indicating that the MR greases based on the new preparation method had good CIP dispersion stability.

### 2.5. Rheological Characterization

The shear stress of MG grease as a function of shear rate for different magnetic fields ranging from 0 to 0.7 T at room temperature was characterized by a parallel-plate magneto-rheometer (Anton Paar, Physica, MCR302, Graz, Austria). The diameter of the test plate was 20 mm and the gap between the parallel plates was 1 mm. The samples of MR greases were filled between the parallel plates. The steady magnetic fields were generated by applying a constant current. The rotational shear experiment was surveyed in the shear rate range of 0.1 to 100 s^−1^. When each test finished, the MR grease samples were degaussed and a new magnetic field was applied [31]. Through the Bingham viscoplastic model, the dynamic yield stress values for the MR grease samples were obtained by calculating shear stress versus shear rate curves to zero shear rate and searching for the point of intersection with the vertical axis [11,18], as show in Figure 3.

## 3. Results and Discussions

Based on the L_9_ orthogonal array test program, nine MR grease samples were successively tested and each response parameter result was obtained under each condition of input parameters (each corresponding row), as shown in Table 2 and Table 3.

Table 2 and Table 3 show that the field-dependent dynamic yield stress for each MR grease sample increased with increasing magnetic field strength. This phenomenon is related to the stronger dipole–dipole interaction between CIPs through the increment of magnetic field strength [8,11]. In the absence of a magnetic field, the dynamic yield stress ranged from 0.215 to 2.506 kPa for all the samples due to the viscoplasticity of the grease medium. The yield stress varied from 26.86 to 65.96 kPa for all MR greases at the maximum magnetic field strength of 0.7041 T. Of all the samples, the eighth MR grease (8#MRG) produced the highest yield stress of 65.96 kPa at the magnetic field strength of 0.7041 T under the input parameter conditions of CIPs fraction—75 wt%, SOV—500 m^2^·s^−1^, CIP size—3.2 μm. This high field-induced yield stress suggested that more stable magnetic particle chain structures were formed at high magnetic field strengths [18].

### 3.1. Qualitative Significance Evaluation of SOV, CIPs Fraction, and CIP Size towards Yield Stress

The analysis of variance (ANOVA) was conducted from the data of Table 2 and Table 3, and the statistical results were obtained, as shown in Table 4, by using MINITAB17 statistical software. The ANOVA based on the Taguchi L_9_(3^3^) test method was applied to confirm the significance of the regression model and input parameters [27,32]. Significant influence of an input parameter on the response parameter can be determined when the F-value for the input parameter, at the 95% confidence level, is more than the critical F-value at the 95% confidence level [32]. In other words, P less than 0.05 indicates a significant effect on the response parameter [27]. Furthermore, it is only when the regression model is significant in relation to the response parameter that the significance analysis of the input parameter (variable) towards the response parameter is meaningful. As shown in Table 4, the regression model for each response parameter included three first-order variables, i.e., the CIPs fraction, SOV, and CIP size. The P-values of the regression model towards all the response parameters (the yield stress under magnetic field strengths in the range 0–0.7041 T) were lower than 0.05. Moreover, the calculated F-ratios for all the regression models were 20.5, 40.95, 31.78, 21.49, 31.56, 33.4, 30.88, and 30.72. These F-ratios were larger than the critical ratio of F at the 95% confidence level, namely F_0.05(3,5)_. Thus, the ternary first order regression model was significant for all the response parameters.

From Table 4, one can see that the CIPs fraction and SOV, with P-values less than 0.05, significantly affect the yields stress under the application of magnetic field strengths ranging from 0 to 0.7041 T. The same result, that CIPs fraction has an important influence on the on-state yield stress, was confirmed in the MR fluids [27]. CIP size obviously affected the yield stress when applying magnetic field strengths varying from 0–0.1306 T. However, the opposite results were produced under the presence of magnetic fields more than 0.1306 T. Larger CIP sizes in the range of 4–53 μm does not influence magnetic field-dependent performance [16]. Different from the CIPs fraction and SOV, the effect of the CIP size on the yield stress generated transformed from significant to insignificant with variation of magnetic field strength. This interesting phenomenon may be related to variation of CIP size resulting from the alteration of the interparticle contact area and suspension microstructure. Microstructural alterations might change the thixotropy and the yield stress of the suspension under low magnetic field strengths [5,19,28].

### 3.2. Quantitative Evaluation for Strength Effects of SOV, CIPs Fraction and Size on Yield Stress

Based on the above significance evaluation, the effects of these input parameters on the response parameter can be qualitatively described. In order to more clearly understand the strength effects of the SOV, CIPs fraction, and CIP size on the field-induced yield stress, it was necessary to quantitatively evaluate the percent contribution (PC) of these input parameters towards the response parameter based on ANOVA. ANOVA can calculate the ratio of the individual sum of squares of a specific variable to the total sum of squares of all the variables [33]. This ratio is the PC of the variable (i.e., each input parameter or error) on the specific response parameter [27,33]. Meanwhile, if the PC of the error is lower than 15%, it is assumed that input parameters of significant influence were considered [34]. As shown in Table 4, the PC of the error varied from 3.911% to 7.519%; thus, the PC of the SOV, CIPs fraction, and CIP size can be used to quantitatively analyze the strength effects of these input parameters on the response parameter. The results of PC for the SOV, CIPs fraction, and CIP size obtained under the influence of different applied magnetic fields are presented in Figure 4. Further analysis showed that the PC of CIPs fraction was dependent on the magnetic field strength. This appears to be due to the stronger interaction induced between the CIPs, which generated thicker magnetic particle chains, resulting in larger flow resistance when the magnetic field strength was increased. However, the PC of SOV exhibited a negative relationship with magnetic field strength, and its PC was lower than the PC of CIPs fraction. This result could be attributed to the relation between the viscous force of the medium and the magnetic forces of particles upon magnetic field application [35]. This indicates that field-induced particle chain structures generate higher repulsive forces compared to viscous force along the direction of the magnetic field. Furthermore, the magnetic forces were significantly increased with increasing magnetic field strength until soft magnetic particles reached saturation. In contrast, the viscous forces were gradually reduced by increasing the magnetic forces.

In addition, Figure 4 shows that the PCs of SOV and CIPs fraction were significantly higher than that of CIP size. The ratio of the PC of CIPs fraction to the PC of CIPs size was more than 8:1 under magnetic fields in the range of 0.1306–0.7041 T. Meanwhile, there were more than triple ratios of the PC of SOV towards the PC of CIP size in the presence of magnetic fields. On top of that, the ratios of the total sum of the PC of the SOV and CIPs fraction to the PC of the CIP size were larger than 11:1 with variation of magnetic field strength. Thus it can be concluded the CIP size ranging from 3.2 μm to 3.9 μm does not significantly influence the on-state yield stress of MR fluids [27].

### 3.3. Direction of The Discrepancy and Constitutive Relation Characterization between SOV, CIPs Fraction, and Yield Stress Based on the Regression Equations

It can be verified that the dynamic yield stress was mainly affected by the SOV and CIPs fraction based on the qualitative and quantitative estimation of the effects of these input parameters on the response parameter. However, CIP size can be ignored since it hardly influences the field-dependent rheological performance of MR grease [16]. On the basis of the ANOVA, multivariate regression equations were developed to characterize the relationships between SOV and CIPs fraction and the yield stress under different magnetic fields. The insignificant term (i.e., CIP size) has been removed from the regression equations. *X*_i_ (i = 1, 2) are the input parameters and subscript i denotes the serial number for the input parameters. *Y*_j_ (j = 0, 1, …, 7) are the response parameters and subscript j denotes the serial number of the response parameters. The binary regression equations for *Y*_0_, *Y*_1_, *Y*_2_, *Y*_3_, *Y*_4_, *Y*_5_, *Y*_6_, *Y*_7_ are:(1)$$Y_{0}=1.198+0.704X_{1}+0.573X_{2}$$
(2)$$Y_{1}=4.115+1.381X_{1}+0.882X_{2}$$
(3)$$Y_{2}=8.307+2.542X_{1}+1.394X_{2}$$
(4)$$Y_{3}=13.533+3.809X_{1}+2.016X_{2}$$
(5)$$Y_{4}=19.347+5.578X_{1}+2.803X_{2}$$
(6)$$Y_{5}=28.938+8.862X_{1}+4.325X_{2}$$
(7)$$Y_{6}=35.1+11.33X_{1}+5.13X_{2}$$
(8)$$Y_{7}=42.75+14.82X_{1}+5.81X_{2}$$

The details of the input parameters (*X*_1_ and *X*_2_ ranging from −1 to 1) and response parameters (*Y*_0_, *Y*_1_, *Y*_2_, *Y*_3_, *Y*_4_, *Y*_5_, *Y*_6_, *Y*_7_) are shown in Table 2 and Table 3. As shown in regression Equations (1)–(8), the CIPs fraction and SOV were linearly related to the yield stress under the influence of different magnetic fields. There is a method to assess the direction of the discrepancy between input parameters in relation to the response parameter by calculating the difference of regression-equation coefficients for the CIPs fraction and SOV [36]. However, this method requires determination of the significance of the different of regression coefficients and is not convenient visual observation [36]. Response surface analysis (RSA) can provide three-dimensional visualization to directly observe the direction of the discrepancy [37]. Thus, RSA was used to analyze the influence direction of the CIPs fraction and SOV versus the yield stress. Based upon regression Equations (1)–(8), each response surface graph was generated in three-dimensional space by using MATLAB2016a software, as shown in Figure 5a–h.

Figure 5a shows the direction of the discrepancy between CIPs fraction and SOV and yield stress in the absence of a magnetic field with variations of the input parameters from the low level (−1) to the high level (1). The yield stress can be enhanced by increasing the CIPs fraction and SOV under no magnetic field. The results indicated the CIPs fraction and SOV generated a synergistic effect on the yield stress in the MR grease. The grease medium was thickened with the increment of SOV, and thus caused an increase in the viscosity of the medium. The increase in viscosity and increase of flow resistance of the suspension was caused by physicochemical interactions leading to the formation of a thicker fiber structure between the medium and CIPs [19,28,38,39]. Thus, the yield stress was strengthened by the production of denser fibrous structure due to the increase of SOV. Meanwhile, similar to MR gels [40], thickening of the grease medium’s viscosity and increment of the CIPs fraction can increase the interparticle friction and viscous forces between the CIPs and the medium, which lead to larger dynamic yield stress.

As in the absence of magnetic fields, Figure 5b–h show that there is the same direction of discrepancy between the input parameters and response parameters in the presence of a magnetic field. The rheological mechanisms were similar but not the same at zero and nonvanishing magnetic fields, although the direction of influence of CIPs fraction and SOV to the response parameter remained consistent throughout. With no external magnetic field, CIPs do not have a permanent magnetic dipole moment [8]. Thus, the CIPs homogeneously dispersed into the grease, forming no particle aggregates [5,18]. Unlike at zero magnetic field, the magnetic dipole moment can be induced in the CIPs under external magnetic fields [18]. As the CIPs fraction is increased, the microstructure of the produced suspension chains to the thick columns under external magnetic fields [8,40]. It was indicated that higher force was required to destroy the magnetic particle chains with the increase of CIPs fraction. Consequently, increasing CIPs fraction and SOV caused formation of a thicker, more cross-linked structure of the magnetic dipole chains and a denser fiber structure of the grease, which together led to increased magnetic field-induced yield stress.

The direction of the discrepancy between the input parameters and response parameter has been investigated through the multivariate regression equation based on ANOVA. Many rheological constitutive models for MR grease have showed the relation between yield stress and variables such as magnetic field strength and CIPs fraction in previous literature [11,24,41]. However, few investigations have queried the mechanism of the influence of CIPs fraction and SOV on field-dependent yield stress. In order to reveal this mechanism, a constitutive model based on the multivariate regression equation was developed to describe the yield stress as a function of the SOV and fraction of CIPs under various magnetic fields. According to regression Equations (1)–(8), they can be initially modeled as shown in Equation (9).
(9)$$Y=n+n_{\emptyset }X_{1}+n_{\eta }X_{2}$$
where *Y* is the field-induced yield stress. *X*_1_ and *X*_2_, ranging from −1 to 1, are the CIPs fraction and SOV, respectively. n is the yield stress coefficient. $n_{\emptyset }$ is the coefficient for CIPs fraction. $n_{\eta }$ is the coefficient for SOV. In Equation (10), dimensionless conversion was performed by encoding the actual values of the CIPs fraction and SOV into *X*_1_ and *X*_2_ [42]. It was necessary to enter the corresponding code for the CIPs and SOV to multivariate regression Equation (9) to calculate the yield stress. However, this equation cannot reveal yield stress as a function of actual values of CIPs fraction and SOV. Thus, it is important to transform the code values (dimensionless parameters) to actual values (dimension parameters) for the CIPs fraction and SOV, as shown in Equations (10) and (11).
(10)$$X_{1}=\frac{\emptyset -\emptyset_{0}}{\max\emptyset -\emptyset_{0}}$$
(11)$$X_{2}=\frac{\eta -\eta_{0}}{\max\eta -\eta_{0}}$$
where *X*_1_ and *X*_2_, ranging from −1 to 1, are dimensionless values for CIPs fraction and SOV, respectively. ∅, ranging from 65 wt% to 75 wt%, is the actual value of CIPs fraction. $\emptyset_{0}$, the actual value for CIPs fraction at the central level, used 0.7 to mean 70 wt% in this study. max∅, the actual value for CIPs fraction at the high level, used 0.75 as 75 wt%. η, ranging from 50 m^2^·s^−1^ to 1000 m^2^·s^−1^, is the actual value of SOV. $\eta_{0}$, the actual value for SOV at the central level, is 500 m^2^·s^−1^. maxη, the actual value for SOV at the high level, is 1000 m^2^·s^−1^. Based upon Equations (9)–(11) and the actual values for the $\emptyset_{0}$, max∅, η, and maxη, the function of yield stress using actual values of CIPs fraction and SOV is characterized as shown in Equation (12).
(12)$$\tau_{y}=n+n_{\emptyset }\left(\frac{\emptyset -0.7}{0.05}\right)+n_{\eta }\left(\frac{\eta -500}{500}\right)$$
$\tau_{y}$ is the field-induced yield stress for Equation (12), namely the *Y* in the Equation (9). However, Equation (12) cannot describe the relationship between yield stress and magnetic field strength. It is easy to observe that the coefficients n, $n_{\emptyset }$, and $n_{\eta }$ monotonically increase with increasing magnetic field, as shown in Table 5. The results indicate the three coefficients were all dependent on the magnetic field strength. Hence, each coefficient as a function of the magnetic field can be characterized by fitting the data of Table 5. The yield stress coefficient, CIPs fraction coefficient, and SOV coefficient as functions of magnetic field strength are shown in Equations (13)–(15), respectively.
(13)$$n\left(B\right)=1.173+8.871B+119.4B^{2}-68.39B^{3}$$
(14)$$n_{\emptyset }\left(B\right)=0.7114+2.123B+25.46B^{2}$$
(15)$$n_{\eta }\left(B\right)=0.6122-1.567B+25.93B^{2}-18.6B^{3}$$
n(B) is the field-induced coefficient of yield stress. $n_{\emptyset }\left(B\right)$ is the field-induced coefficient of CIPs fraction. $n_{\eta }\left(B\right)$ is the field-induced coefficient of SOV. The values of determination $R_{r}^{2}$ for magnetic field versus each field-induced coefficient were calculated. $R_{r}^{2}$ for the coefficient of yield stress, coefficient of CIPs fraction, and coefficient of SOV were $R_{r-n\left(B\right)}^{2}=100\%$, $R_{r-n_{\emptyset }\left(B\right)}^{2}=99.99\%$ and $R_{r-n_{\eta }\left(B\right)}^{2}=99.81\%$, respectively. Finally, a new dynamic yield stress model, in which the yield stress is a function of CIPs fraction, SOV, and magnetic field strength, was developed on the basis of a multivariate regression Equation. The yield stress model is shown in Equation (16), which was obtained by substituting Equations (13)–(15) into Equation (12).
(16)$$\tau_{y}=n\left(B\right)+n_{\emptyset }\left(B\right)\cdot \left(20\emptyset -14\right)+n_{\eta }\left(B\right)\left(0.002\eta -1\right)$$

The yield stress model was verified by comparing with experimental $\tau_{y}-B$ for all the MR grease samples. Comparison plots are shown in Figure 6a–f. The accuracy of the yield stress model was quantified by coefficient of determination ($R^{2}$) and the mean squared error (MSE) [43], as shown in Table 6.

As shown in Figure 6a–c, the yield stress was increased with increasing CIPs fraction when the SOV was at a constant value. Figure 6d–f shows that the yield stress increased with increasing SOV when the CIPs fraction was at a constant value. These results were also obtained in the RSA. In addition, Table 6 shows that the coefficient of determination $R^{2}$ for all the MR grease samples was in the range of 93.75–99.27%. This indicates that this yield stress model based on the multivariate regression equation can accurately reveal the constitutive relationship between CIPs fraction and SOV and yield stress under application of different magnetic field strengths.

## 4. Conclusions

In this study, various MR grease samples were prepared using an orthogonal L_9_ array based on a new preparation technique. Experimental tests for the MR grease samples were conducted under different magnetic fields by using rotational rheometry. Four methods, including the ANOVA, comparison of PC, RSA, and a constitutive model, were used to study effects of SOV, CIPs fraction, and CIP size on the field-induced yield stress. The qualitative evaluation has shown that CIPs fraction and SOV influence the field-induced yield stress of MR grease; however, based upon the ANOVA, CIP size has a negligible effect. The quantitative evaluation has shown that field-induced yield stress is primarily affected by CIPs fraction and SOV based on the comparison of PC. RSA showed that there are synergistic effects of CIPs fraction and SOV on the yield stress. The constitutive model with accuracy higher than 93.75%, on the basis of multivariate regression, successfully fitted all the MR grease samples to characterize yield stress as a function of CIPs fraction, SOV, and magnetic field strength. From these results, we conclude that the field-induced yield stress of MR grease is closely related to CIPs fraction and SOV. Furthermore, the highest yield stress of MR grease realized in this research was 65.96 kPa. How other materials, such as thicker agents or additives, affect the field-dependent rheological performance of MR grease and investigation of the microstructure of MR grease using SEM will be conducted as a second stage of this work.

## Figures and Tables

**Figure 1 materials-12-01778-f001:**
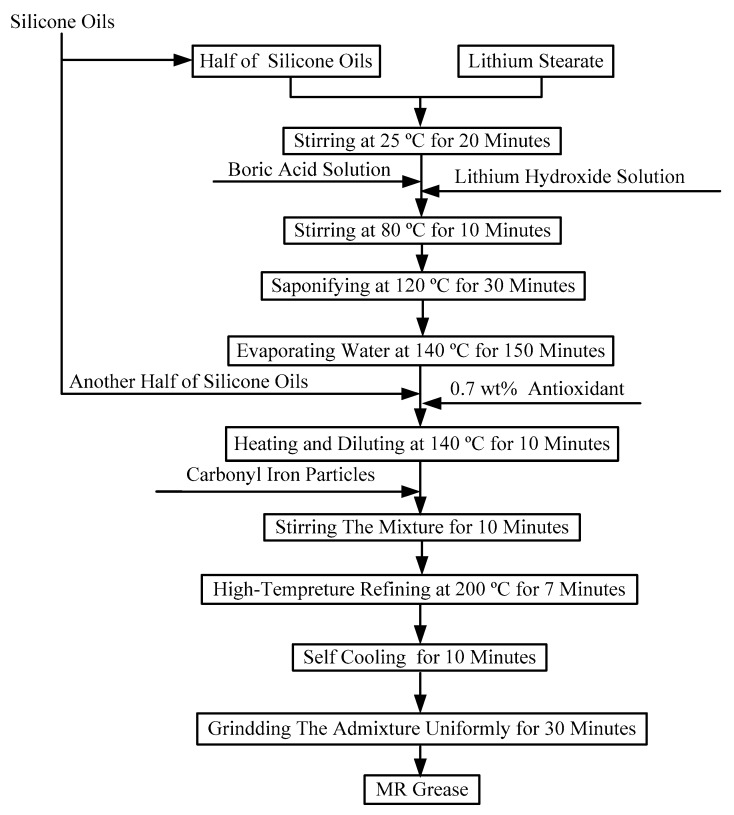
The preparation procedure of magnetorheological (MR) grease.

**Figure 2 materials-12-01778-f002:**
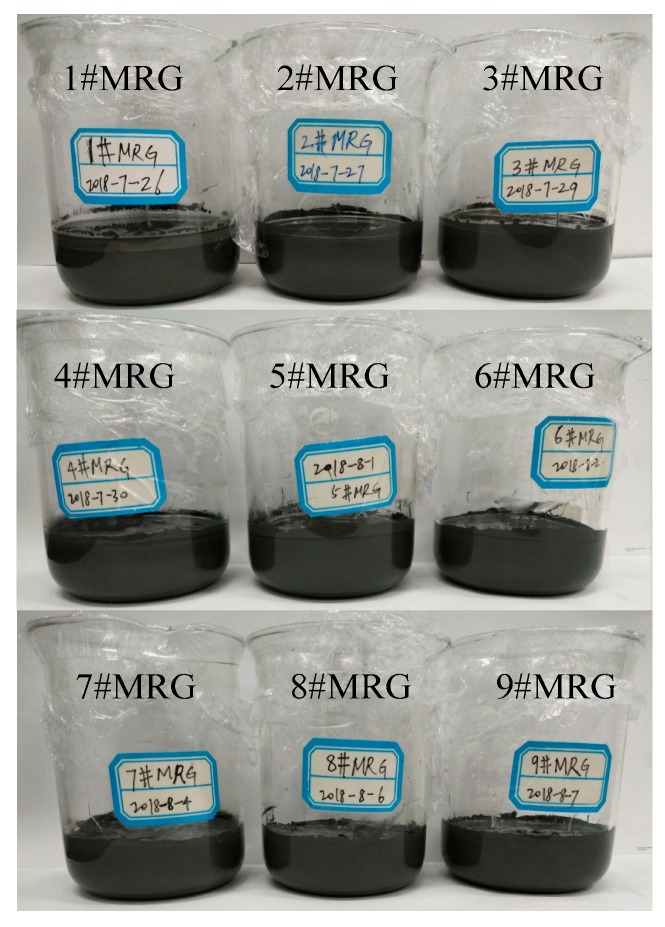
Appearance of MR greases after holding in a static state for 30 days.

**Figure 3 materials-12-01778-f003:**
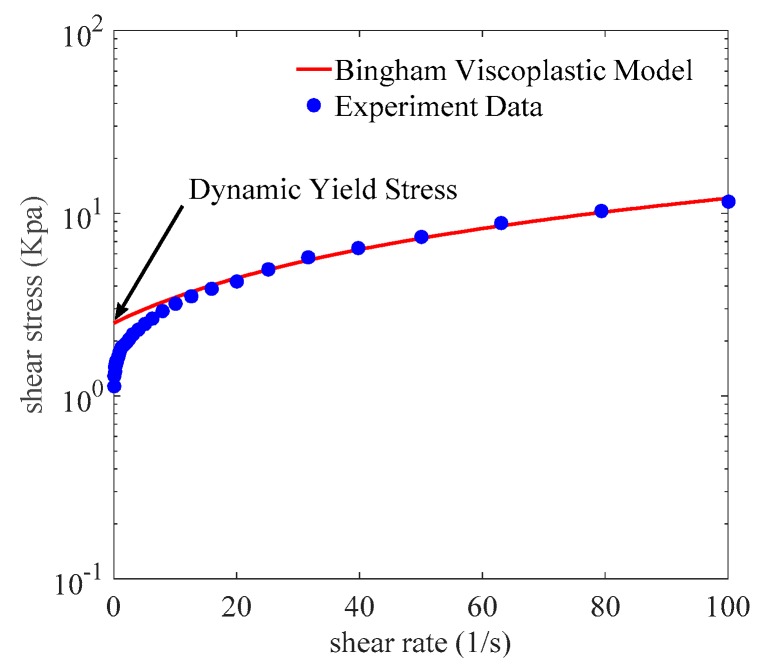
Calculation of dynamic yield stress of MR grease at zero shear rate

**Figure 4 materials-12-01778-f004:**
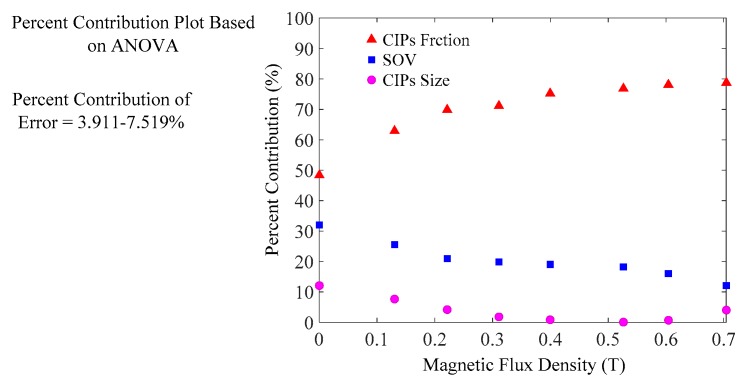
Percent contribution as a function of the SOV, CIPs fraction, and CIP size under different magnetic fields.

**Figure 5 materials-12-01778-f005:**
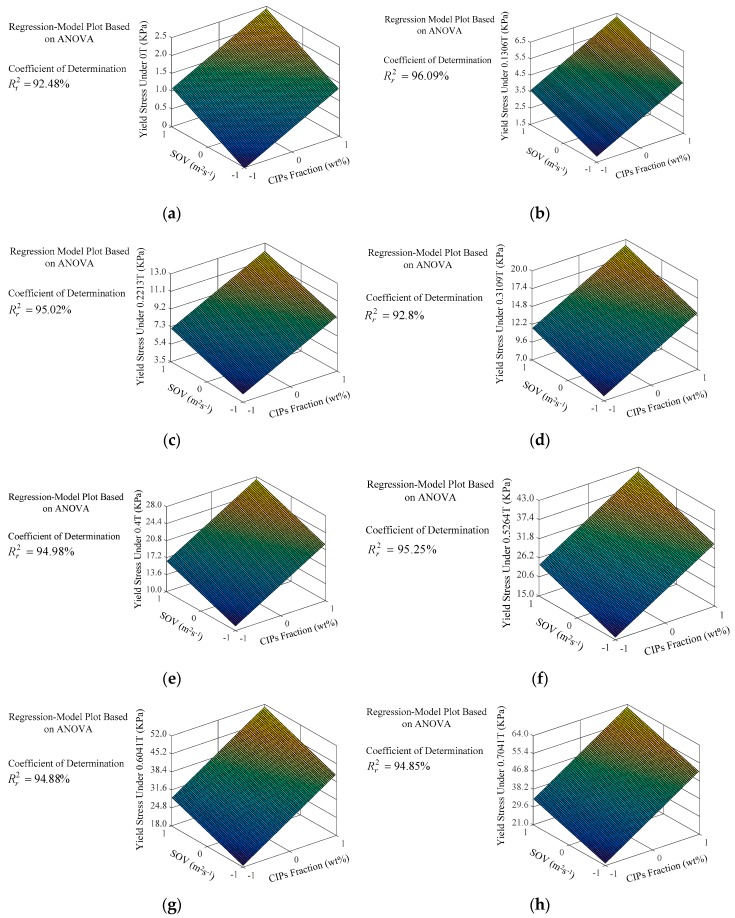
(**a**) Response surface analysis (RSA) of yield stress under a magnetic field of 0 T; (**b**) RSA of yield stress under a magnetic field of 0.1306 T; (**c**) RSA of yield stress under a magnetic field of 0.2213 T; (**d**) RSA of yield stress under a magnetic field of 0.3109 T; (**e**) RSA of yield stress under a magnetic field of 0.4 T; (**f**) RSA of yield stress under a magnetic field of 0.5264 T; (**g**) RSA of yield stress under a magnetic field of 0.6041 T; (**h**) RSA of yield stress under a magnetic field of 0.7041 T.

**Figure 6 materials-12-01778-f006:**
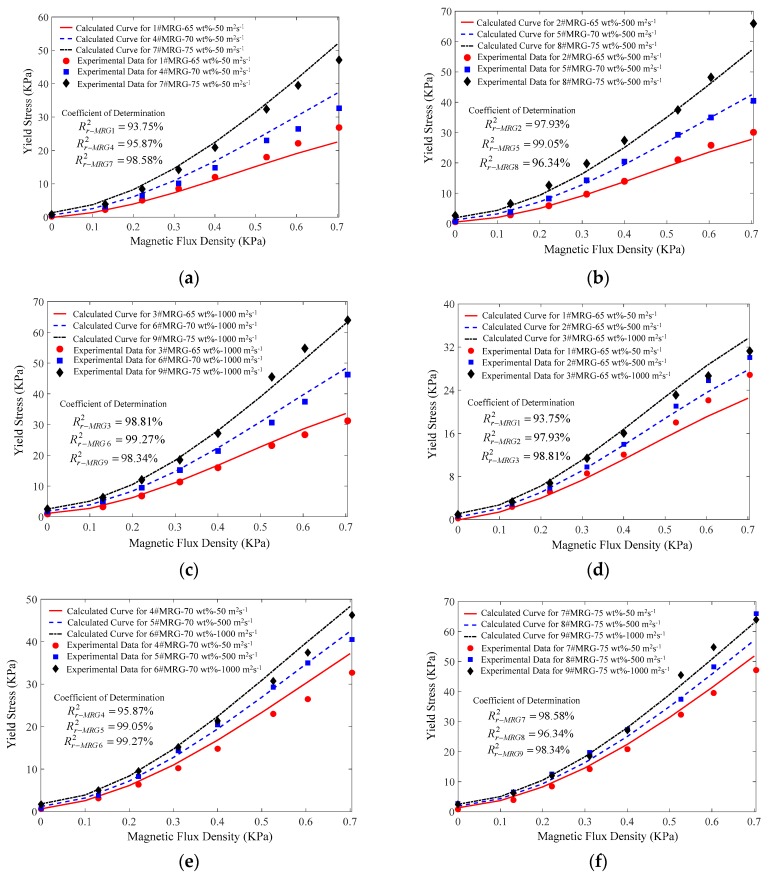
(**a**) Comparison of experimental and predicted yield stress by the proposed model for three samples with CIPs fraction in the range of 65–75 wt% and SOV at 50 m^2^·s^−1^; (**b**) Comparison of experimental and predicted yield stress by the proposed model for three samples with CIPs fraction in the range of 65–75 wt% and SOV at 500 m^2^·s^−1^; (**c**) Comparison of experimental and predicted yield stress by the proposed model for three samples with CIPs fraction in the range of 65–75 wt% and SOV at 1000 m^2^·s^−1^; (**d**) Comparison of experimental and predicted yield stress by the proposed model for three samples with SOV in the range of 50–1000 m^2^·s^−1^ and CIPs fraction at 65 wt%; (**e**) Comparison of experimental and predicted yield stress by the proposed model for three samples with SOV in the range of 50–1000 m^2^·s^−1^ and CIPs fraction at 70 wt%; (**f**) Comparison of experimental and predicted yield stress by the proposed model for three samples with SOV in the range of 50–1000 m^2^·s^−1^ and CIPs fraction at 75 wt%.

**Table 1 materials-12-01778-t001:** Input parameters and their levels. CIPs—carbonyl iron particles; SOV—silicone oil viscosity.

Coding	Input Parameters and Their Levels		
	CIPs Fraction, X_1_ (wt%)	SOV, X_2_ (m^2^·s^−1^)	CIPs Size, X_3_ (μm)
−1	65	50	3.2
0	70	500	3.5
1	75	1000	3.9

**Table 2 materials-12-01778-t002:** Experimental results.

Samples	Input Parameters			Response Parameters			
	CIPs Fraction, *X*_1_ (wt%)	SOV, *X*_2_ (m^2^·s^−1^)	CIP Size, *X*_3_ (μm)	Yield Stress Under 0 T, *Y*_0_ (kPa)	Yield Stress Under 0.1306 T, *Y*_1_ (kPa)	Yield Stress Under 0.2213 T, *Y*_2_ (kPa)	Yield Stress Under 0.3109 T, *Y*_3_ (kPa)
1#MRG	−1	−1	−1	0.215	2.314	5.108	8.575
2#MRG	−1	0	0	0.648	2.828	5.846	9.742
3#MRG	−1	1	1	0.896	3.29	6.772	11.33
4#MRG	0	−1	0	0.618	3.1	6.381	10.19
5#MRG	0	0	1	0.739	3.797	8.211	14.29
6#MRG	0	1	−1	1.681	4.99	9.466	15.17
7#MRG	1	−1	1	0.812	3.874	8.405	14.2
8#MRG	1	0	−1	2.663	6.546	12.55	19.74
9#MRG	1	1	0	2.506	6.3	12.02	18.56

**Table 3 materials-12-01778-t003:** Experimental results.

Samples	Input Parameters			Response Parameters			
	CIPs Fraction, *X*_1_ (wt%)	SOV, *X*_2_ (m^2^·s^−1^)	CIP Size, *X*_3_ (μm)	Yield Stress Under 0.4 T, *Y*_4_ (kPa)	Yield Stress Under 0.5264 T, *Y*_5_ (kPa)	Yield Stress Under 0.6041 T, *Y*_6_ (kPa)	Yield Stress Under 0.7041 T, *Y*_7_ (kPa)
1#MRG	−1	−1	−1	12.04	17.98	22.1	26.86
2#MRG	−1	0	0	13.99	21.03	25.77	30.08
3#MRG	−1	1	1	15.99	23.12	26.67	31.25
4#MRG	0	−1	0	14.79	23.02	26.48	32.67
5#MRG	0	0	1	20.46	29.29	34.95	40.49
6#MRG	0	1	−1	21.36	30.7	37.42	46.25
7#MRG	1	−1	1	20.89	32.36	39.5	47.14
8#MRG	1	0	−1	27.41	37.45	48.27	65.96
9#MRG	1	1	0	27.19	45.49	54.76	64.02

**Table 4 materials-12-01778-t004:** Analysis of variance (ANOVA) for response parameters.

Response Parameters	Source	SS	DOF	MS	Contribution	F	*p*
Yield Stress Under 0 T	Regression Model	5.6843	3	1.89476		20.50	0.003
	*X*_1_-CIPs Fraction	2.9709	1	2.97088	48.336%	32.14	0.002
	*X*_2_-SOV	1.9700	1	1.96997	32.050%	21.31	0.006
	*X*_3_-CIPs Size	0.7434	1	0.74342	12.095%	8.04	0.036
	Error	0.4622	5	0.09243	7.519%		
	Total	6.1464	8		100.000%		
Yield Stress Under 0.1306 T	Regression Model	17.5071	3	5.8357		40.95	0.001
	*X*_1_-CIPs Fraction	11.4485	1	11.4485	62.836%	80.34	<0.001
	*X*_2_-SOV	4.6675	1	4.6675	25.618%	32.75	0.002
	*X*_3_-CIPs Size	1.3911	1	1.3911	7.635%	9.76	0.026
	Error	0.7125	5	0.1425	3.911%		
	Total	18.2196	8		100.000%		
Yield Stress Under 0.2213 T	Regression Model	52.741	3	17.5803		31.78	0.001
	*X*_1_-CIPs Fraction	38.755	1	38.7553	69.820%	70.05	<0.001
	*X*_2_-SOV	11.659	1	11.6594	21.005%	21.07	0.006
	*X*_3_-CIPs Size	2.326	1	2.3263	4.190%	4.20	0.096
	Error	2.766	5	0.5533	4.985%		
	Total	55.507	8		100.000%		
Yield Stress Under 0.3109 T	Regression Model	113.663	3	37.888		21.49	0.003
	*X*_1_-CIPs Fraction	87.043	1	87.043	71.068%	49.38	0.001
	*X*_2_-SOV	24.382	1	24.382	19.907%	13.83	0.014
	*X*_3_-CIPs Size	2.239	1	2.239	1.828%	1.27	0.311
	Error	8.814	5	1.763	7.197%		
	Total	122.478	8		100.000%		
Yield Stress Under 0.4 T	Regression Model	235.866	3	78.622		31.56	0.001
	*X*_1_-CIPs Fraction	186.707	1	186.707	75.187%	74.95	<0.001
	*X*_2_-SOV	47.152	1	47.152	18.988%	18.93	0.007
	*X*_3_-CIPs Size	2.007	1	2.007	0.808%	0.81	0.411
	Error	12.456	5	2.491	5.017%		
	Total	248.322	8		100.000%		
Yield Stress Under 0.5264 T	Regression Model	583.717	3	194.572		33.40	0.001
	*X*_1_-CIPs Fraction	471.175	1	471.175	76.883%	80.87	<0.001
	*X*_2_-SOV	112.234	1	112.234	18.314%	19.26	0.007
	*X*_3_-CIPs Size	0.308	1	0.308	0.050%	0.05	0.827
	Error	29.131	5	5.826	4.753%		
	Total	612.848	8		100.000%		
Yield Stress Under 0.6041 T	Regression Model	935.654	3	311.885		30.88	0.001
	*X*_1_-CIPs Fraction	770.440	1	770.440	78.126%	76.29	<0.001
	*X*_2_-SOV	157.799	1	157.799	16.002%	15.63	0.011
	*X*_3_-CIPs Size	7.415	1	7.415	0.752%	0.73	0.431
	Error	50.494	5	10.099	5.120%		
	Total	986.148	8		100.000%		
Yield Stress Under 0.7041 T	Regression Model	1588.45	3	529.48		30.72	0.001
	*X*_1_-CIPs Fraction	1318.09	1	1318.09	78.710%	76.48	<0.001
	*X*_2_-SOV	202.42	1	202.42	12.088%	11.74	0.019
	*X*_3_-CIPs Size	67.94	1	67.94	4.057%	3.94	0.104
	Error	86.17	5	17.23	5.145%		
	Total	1674.62	8		100.000%		

DOF = Degree of Freedom, SS = Sum of Square, MS = Mean Sum of Squares, F_0.05(3,5)_ = 5.41.

**Table 5 materials-12-01778-t005:** Key parameters of MR grease under different magnetic field strengths.

Magnetic Field, B/T	Yield Stress Coefficient, *n*/kPa	CIPs Fraction Coefficient, *n*_∅_/(kPa/wt%)	SOV Coefficient, *n_η_*/(kPa/m^2^·s^−1^)
0	1.198	0.704	0.573
0.1306	4.115	1.381	0.882
0.2213	8.307	2.542	1.394
0.3109	13.533	3.809	2.016
0.4	19.347	5.578	2.803
0.5264	28.938	8.862	4.325
0.6041	35.1	11.33	5.13
0.7041	42.75	14.82	5.81

**Table 6 materials-12-01778-t006:** Model accuracy in terms of $R^{2}$ and MSE.

Accuracy Evaluation	*R* ^2^	MSE
1#MRG	93.75%	7.7984
2#MRG	97.93%	9.4778
3#MRG	98.81%	11.3449
4#MRG	95.87%	12.4756
5#MRG	99.05%	14.1563
6#MRG	99.27%	16.0243
7#MRG	98.58%	17.1744
8#MRG	96.34%	18.8538
9#MRG	98.34%	20.7205

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
