# Peer review of "Effects of Silicone Oil Viscosity and Carbonyl Iron Particle Weight Fraction and Size on Yield Stress for Magnetorheological Grease Based on a New Preparation Technique"

_materials, 2019, doi:10.3390/ma12111778_

Round 1

Reviewer 1 Report

Thanks for that interesting paper.

please could you use Si-unit: not cSt---> m2s-1

at page 2 line 48 : you describe greases as base oil + soap thickener. Not all thickener are soaps or metal soap. Please can you write: thickener with different materials

page 6 from line 212: I think for the reader of this journal it could be helpful to present a diagram (as you described) to understand the way you get the yield stress

I was a little bit confused because of table 2. You use two names. Please can you check it

Author Response

Response to Reviewer 1 Comments

Point 1: 
please could you use Si-unit: not cSt---> m2s-1

Response 1: Thanks for the reviewer’s comment. According to the reviewer’s comment, we have replaced the unit for the viscosity ‘cSt’ with the Si-unit ‘m2s-1’ in the paper as shown in page 1 line 17, page 4 line 156, page 7 line 234, page 12 lines 407-409, page 14 lines 444-451, table 1, table 2, table 4, figures 5a-h and figures 6a-f.

Point 2: at page 2 line 48 : you describe greases as base oil + soap thickener. Not all thickener are soaps or metal soap. Please can you write: thickener with different materials

Response 2: Thank the reviewer’s insightful comment. Based on the comment of reviewer, we have used the ‘thickener with different materials’ to replace ‘a soap thickener’ for precisely describing the thickener as shown in page 2 line 48.

Point 3: page 6 from line 212: I think for the reader of this journal it could be helpful to present a diagram (as you described) to understand the way you get the yield stress

Response 3: Thank the reviewer’s suggestion. In order to be helpful to understand the way we get the yield stress based upon the reviewer’s comment, the authors have added a diagram to present the calculated way for dynamic yield stress of MR grease at zero shear rate as shown in Figure 3.

Point 4: I was a little bit confused because of table 2. You use two names. Please can you check it

Response 4: Thank the reviewer’s comment. We have used the same name for table 2 with the title ‘Experimental results’. 

Reviewer 2 Report

the manuscript proposed by Wang et al. talk about the magnetorheological performance of silicone oil viscosity with  carbonyl iron particles. I think that the paper, if improved, can be accepted from the journal.

-please re-phrase the sentence in line 145, "in [11]"

-into the manuscript is report the word L9 in two different ways: "L9" and "L9" please harmonise. 

-the list of the Y parameters "Y0, Y1, Y2, Y3, Y4, Y5, Y6, Y7" can be removed

-the paragraph 3.3 and 3.4 get me confused. I suggest at the authors to merge them and delete the unnecessary part, remember you that the manuscript is an article and not a thesis.

Author Response

Response to Reviewer 2 Comments

Point 1: 
Please re-phrase the sentence in line 145, "in [11]"

Response 1: Thanks for the reviewer’s comment. We have replaced ‘In [11], it was claimed’ with ‘Park et al [11] had found’ as shown in page 4 line 148.

Point 2: Into the manuscript is report the word L9 in two different ways: "L9" and "L9" please harmonise.

Response 2: Thank the reviewer’s insightful comment. We have used the "L9" to replace the "L9" as shown in page 7 line 240 and page 15 line 463.

Point 3: The list of the Y parameters "Y0, Y1, Y2, Y3, Y4, Y5, Y6, Y7" can be removed

Response 3: Thank the reviewer’s comment. We have removed the Y parameters "Y0, Y1, Y2, Y3, Y4, Y5, Y6, Y7" in the table 3.

Point 4: The paragraph 3.3 and 3.4 get me confused. I suggest at the authors to merge them and delete the unnecessary part, remember you that the manuscript is an article and not a thesis.

Response 4: Thank the reviewer’s suggestion. Considering the reviewer’s suggestion, we have merged the paragraph 3.4 into the paragraph 3.3 for the revised manuscript. Because the research of paragraph 3.3 were conducted basing on the regression equation, the heading of paragraph 3.3 has been amended as " Direction of The Discrepancy and Constitutive Relation Characterization between SOV, CIPs Fraction to Yield Stress Based on The Regression Equations " as shown in lines 306-307. Besides, we have added the necessary descriptions to connect link between research for direction of the discrepancy and constitutive relation characterization of SOV and CIPs fraction to the yield stress as shown in lines 380-383. Since each part of the paragraph 3.3 is necessary and complete in the paper, we have not conducted to delete contents of the paragraph 3.3.

Reviewer 3 Report

This paper is worth of publication in the journal of Results in Physics, although the following minor changes are still required:

The introduction suffers from lack of enough lit review.

The stats method needs to be added and error bars/ error values are missed in all graphs/figures.

Adding high mag optical microscopy or SEM is very helpful to understand the materials and methodology in the papers.

Author Response

Response to Reviewer 3 Comments

Point 1: 
 The introduction suffers from lack of enough lit review.

Response 1: Thanks for the reviewer’s comment. We have added the review " In brief, the above-mentioned literature analysis shows that some research workers mainly focused on the researches such as how the viscoplastic medium, nanoadditives, shape of magnetic particles, large CIPs size and fraction affect the field-induced rheological behavior for the MR grease basing on a conventional preparation technique." into the introduction as shown in page 3 lines 107-110.

Point 2: The stats method needs to be added and error bars/ error values are missed in all graphs/figures.

Response 2: Thank the reviewer’s insightful comment. We have added the statistic method (i.e. ANVOA) and error values (i.e. percent contribution of error and coefficient of determination) in the related figures as shown in figure 4, figures 5a-h and figures 6a-f.

Point 3: Adding high mag optical microscopy or SEM is very helpful to understand the materials and methodology in the papers.

Response 3: Thank the reviewer’s suggestion. Due to the lack of the optical microscopy and SEM, we are making efforts to seek the collaborators and will finish the investigation for the microstructure of MR grease in the second stage of this work as shown in page 15 line 479.
